# Modeling lung perfusion abnormalities to explain early COVID-19 hypoxemia

Jacob Herrmann [1✉], Vitor Mori [2], Jason H. T. Bates[2] & Béla Suki [1]

Early stages of the novel coronavirus disease (COVID-19) are associated with silent hypoxia and poor oxygenation despite relatively minor parenchymal involvement. Although speculated that such paradoxical findings may be explained by impaired hypoxic pulmonary vasoconstriction in infected lung regions, no studies have determined whether such extreme degrees of perfusion redistribution are physiologically plausible, and increasing attention is directed towards thrombotic microembolism as the underlying cause of hypoxemia. Herein, a mathematical model demonstrates that the large amount of pulmonary venous admixture observed in patients with early COVID-19 can be reasonably explained by a combination of pulmonary embolism, ventilation-perfusion mismatching in the noninjured lung, and normal perfusion of the relatively small fraction of injured lung. Although underlying perfusion heterogeneity exacerbates existing shunt and ventilation-perfusion mismatch in the model, the reported hypoxemia severity in early COVID-19 patients is not replicated without either extensive perfusion defects, severe ventilation-perfusion mismatch, or hyperperfusion of nonoxygenated regions.

[1] Department of Biomedical Engineering, Boston University, Boston, MA, USA. [2] Department of Medicine, University of Vermont, Burlington, VT, USA.
✉email: jakeherr@bu.edu

Early reports of the disease caused by the novel coronavirus (COVID-19) describe at least a subset of patients who present with hypoxemia while breathing room air and show minimal nonaerated lung tissue on chest computed tomographic (CT) imaging, but the frequency of this presentation is uncertain[1,2]. Other reports, however, describe features reminiscent of conventional acute respiratory distress syndrome (ARDS)[2–4], including a progressively increasing fraction of afflicted lung in which poorly aerated lung regions eventually become completely nonaerated[5,6]. The early stages of COVID-19 thus remain poorly understood. Nevertheless, the so-called "silent hypoxia" frequently seen in early COVID-19 patients[7,8] has garnered a great deal of attention recently in both the literature and the press, and begs a physiologic explanation.

An early report in a small COVID-19 cohort estimates an average shunt fraction of 50% with an average fraction of nonaerated lung on CT of only 17%. This gives a ratio of shunt to nonaerated lung of 3, which is large compared with values of $1.25 \pm 0.8$ reported for ARDS[1,9]. A possible interpretation of this finding is that a disproportionately large fraction of the pulmonary circulation in COVID-19 patients is being directed through poorly aerated or nonaerated lung[1,3,10], which has led some to hypothesize an underlying impairment in hypoxic pulmonary vasoconstriction (HPV)[1,11]. That is, as HPV normally involves a local feedback mechanism whereby pulmonary arterioles constrict in response to poor regional oxygenation[12], a failure of this response could potentially lead to a significant mismatch between ventilation and perfusion[13]. Impaired HPV is not the only possible explanation, however. COVID-19 is often associated with coagulopathy[14,15] that can lead to microemboli, which in turn could redirect perfusion to lung regions having low or zero ventilation:perfusion ratios[16,17]. Diffusion limitation that might arise when inflammation and edema thicken the blood–gas barrier can also cause hypoxemia.

Whether any of the above explanations are actually plausible, however, requires a quantitative analysis of the physiologic factors that determine arterial oxygenation. For example, given a certain fraction of injured lung ($F_{inj}$) with impaired oxygen transport, what increase in regional blood flow and hence vasodilation would be necessary to manifest a ratio of shunt fraction ($F_{shu}$) to $F_{inj}$ around 3? Could a ratio of this magnitude be explained by impaired oxygen equilibration? What roles might gravitational gradients and thromboembolic perfusion defects play? In the present study we use a mathematical model of perfusion and oxygen transport to address these questions with the goal of determining if the impaired HPV and microemboli hypotheses can potentially explain the hypoxemia of early COVID-19, or whether we need to look for an alternative explanation.

## Results

**Model overview.** The lung model was partitioned into 12 compartments (Fig. 1), representing three different height levels each partitioned into four compartments: injured-perfused, normal-perfused, injured-nonperfused, and normal-nonperfused. Each perfused compartment received deoxygenated mixed venous blood and returned end-capillary blood with oxygen content determined by injury severity. Perfusion distribution in the model reflected the relative vascular resistance in each compartment as well as the presence of perfusion defects in the nonperfused compartments (e.g., owing to pulmonary embolism). Baseline resistances were determined by a specified baseline perfusion gradient, defined as half the range of perfusion across all height levels divided by the average. Vascular resistance was then adjusted in injured regions to reflect possible abnormalities arising in COVID-19. Three types of modification were

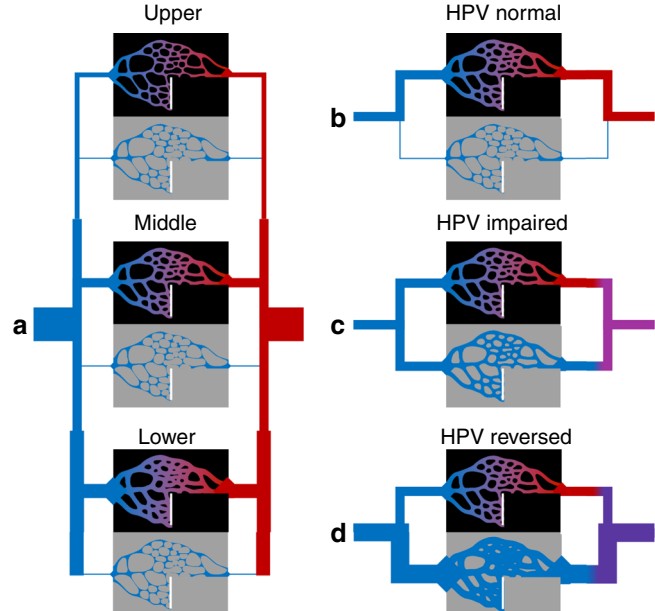

**Fig. 1 Model overview. a** Schematic of the 12-compartment model used to simulate distributed perfusion in aerated and injured compartments at different height levels. Deoxygenated mixed venous blood (blue) passes through aerated (black) or injured (gray) compartments, and returns to the oxygenated mixed arterial blood (red). Vascular resistance in each compartment was determined by height level, presence of perfusion defect or pulmonary embolism (white bars), as well as the degree of oxygenation or injury. Hypoxic pulmonary vasoconstriction (HPV) could be either **b** "normal" with reduced perfusion to regions of low end-capillary oxygen content, **c** "impaired" with no response, or **d** "reversed" with increased perfusion to injured regions.

examined: (1) normal HPV function increased resistance exponentially in regions with low end-capillary oxygen tension[18]; (2) impaired HPV function produced no change in resistance; and (3) reversed HPV-reduced resistance regardless of oxygenation. Measured outcomes included $F_{shu}$, ratio $F_{shu}:F_{inj}$, and ratio arterial oxygen tension to fractional inspired oxygen ($P_aO_2:F_iO_2$). In most cases, model conditions were intended to represent patients upon admission, without supplemental oxygen (i.e., $F_iO_2 = 21\%$), mechanical ventilation, or positive airway pressure. Conditions representing critical care patients are available as supplementary material, with increased $F_{inj}$ (Supplementary Figs. 5–9), increased $F_iO_2$ (Supplementary Fig. 10), and increased mixed venous oxygen tension (e.g., owing to venovenous extracorporeal membrane oxygenation) (Supplementary Fig. 11).

**Vasodilation in shunt regions can explain hypoxemia.** The variability in $F_{shu}:F_{inj}$ with respect to injury location and HPV alterations is shown in Fig. 2. For simplicity, the extent of injury in each simulation is restricted to only one height zone: lower, middle, or upper. Perfusion defect is not considered in this figure. Normal HPV function results in the lowest pulmonary shunt fractions and lowest degree of hypoxemia, preventing $P_aO_2:F_iO_2 < 300$ mmHg until $F_{inj} > 30\%$. Given a relatively small fraction of injured lung, with $F_{inj}$ ranging from 0 to 30%, both a complete shunt (i.e., zero oxygen uptake) and reversal of HPV (i.e., vasodilation in injured regions) are necessary conditions for $F_{shu}:F_{inj} > 2$ and $P_aO_2:F_iO_2 < 300$ mmHg. By contrast, impairment of HPV alone is not sufficient to produce reported levels of severe hypoxemia at low values of $F_{inj}$[1]. With HPV impairment, $F_{shu}$ more closely follows $F_{inj}$ such that the ratio of $F_{shu}$ to $F_{inj}$ lies between 0.7 and 1.3. For all considered alterations to HPV,

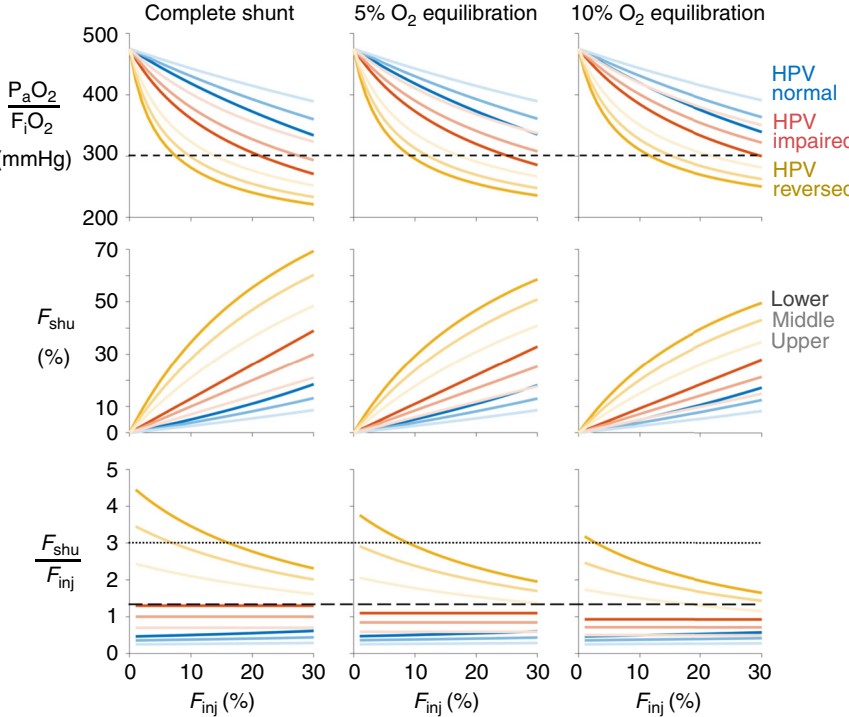

**Fig. 2 Effects of alterations to hypoxic pulmonary vasoconstriction (HPV).** Severity of pulmonary shunt with respect to fractional injury extent ($F_{inj}$), type of HPV modification (blue: normal; red: impaired; yellow: reversed), injury location within one height level (light to dark color). Rows correspond to the ratio of arterial oxygen tension to inspired oxygen fraction ($P_aO_2$:$F_iO_2$), shunt fraction ($F_{shu}$), and ratio of shunt fraction to injured fraction ($F_{shu}$:$F_{inj}$). Columns correspond to varying degrees of impaired oxygen equilibration between capillary blood and alveolar gas in the injured region. Baseline perfusion gradient was 30%, and reversed HPV was modeled with 72% reduction of vascular resistance in injured regions. Short-dashed line in the top row indicates $P_aO_2$: $F_iO_2 = 300$ mmHg, a threshold for ARDS. Dotted and long-dashed lines in the bottom row indicate $F_{shu}$:$F_{inj}$ ratios of 3.0 and 1.3, respectively, values reported for COVID-19 and ARDS patients, respectively[1].

focusing the injury in the lower zone (i.e., those with higher baseline perfusion) results in higher $F_{shu}$ and worse hypoxemia. Note that the model does not exhibit $P_aO_2$:$F_iO_2 < 190$ mmHg, owing to the selection of 21% inspired oxygen and assumption of 40 mmHg mixed venous oxygen tension (40/0.21 = 190.5). Lower $P_aO_2$:$F_iO_2$ ratios are expected in patients with pulmonary shunt when higher $F_iO_2$ is administered; however, the $F_{shu}$:$F_{inj}$ ratio is insensitive to increasing $F_iO_2$, especially if there is complete shunt in the injured regions (Supplementary Fig. 10). Hypoxia is often associated with mixed venous oxygen tensions below 40 mmHg, which can also yield lower $P_aO_2$:$F_iO_2$ ratios but still do not affect $F_{shu}$, except in the case of normal HPV function (Supplementary Fig. 11). Interestingly, as $F_{inj}$ decreases in the reversed HPV model, the $F_{shu}$:$F_{inj}$ ratio increases, indicating that $F_{shu}$ decreases more slowly than $F_{inj}$. Note that the impaired HPV model represents unaltered vascular resistances from baseline values, and therefore corresponds to a model with relatively uniform perfusion distribution.

**Perfusion gradients can exacerbate apparent venous admixture.** The interplay between baseline perfusion gradients and vasodilation in the HPV reversal model is shown in Fig. 3. Again, perfusion defect is not considered in this figure. Baseline perfusion gradient varies between 0% and 100%, representing a range of perfusion heterogeneity from uniform with 1/3-1/3-1/3 distribution at 0% gradient to 0-1/3-2/3 distribution at 100% gradient. Pulmonary shunt and hypoxemia both become more severe with increases in either the vasodilation (i.e., reduction in vascular resistance) of injured regions as a result of reversed HPV or the baseline perfusion gradient. Both of these factors determine the

overall degree of perfusion heterogeneity in the injured lung, and in the specific case of injury focused in the lower lung, both contribute to enhanced perfusion to the injured region. Hypoxemia and shunt are more more sensitive to the degree of vasodilation compared to the baseline gradient. The ratio of $F_{shu}$:$F_{inj} = 3$ is represented in each panel by contours of $F_{shu}$ at 30%, 60%, and 90% for $F_{inj}$ at 10%, 20%, and 30%, respectively. Note that baseline perfusion gradient does not explicitly require the definition of upright vs. supine vs. prone positioning, but instead simply reflects discrepant perfusion in three arbitrary lung compartments.

**Extensive perfusion defect can also explain hypoxemia.** The role of perfusion defect and pulmonary emboli is shown in Fig. 4, demonstrating the amount of perfusion defect required to produce a ratio $F_{shu}$:$F_{inj} \geq 3$ for each type of altered HPV function. Corresponding results for arterial oxygen saturation and oxygen tension are provided in Supplementary Figs. 1–4. When pulmonary emboli are located only in the injured compartments, or present in both normal and injured compartments with equal probabilities, no amount of perfusion defect is sufficient to produce $F_{shu}$:$F_{inj} \geq 3$ in the models of either normal or impaired HPV function. Ratios of $F_{shu}$:$F_{inj} \geq 3$ are observed only when extensive perfusion defect (i.e., creating >60% perfusion defect) is concentrated within the noninjured compartments. By contrast, the model of reversed HPV function can exhibit $F_{shu}$:$F_{inj} \geq 3$ without any pulmonary embolism. Increasing perfusion defect has no effect on $F_{shu}$:$F_{inj}$ when pulmonary emboli are uniformly distributed, but exacerbates $F_{shu}$:$F_{inj}$ when focused within the noninjured lung. Figure 4 demonstrates that increasing the size of the

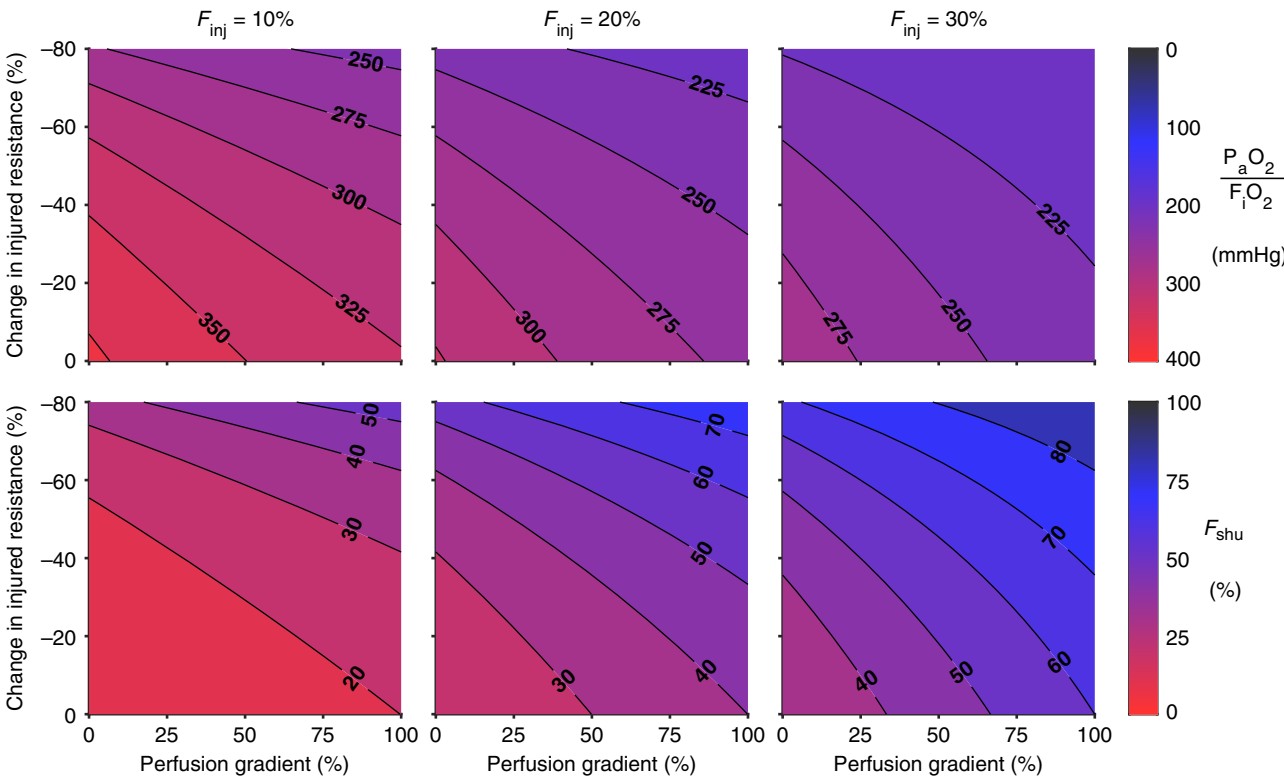

**Fig. 3 Hypoxemia worsened by increased perfusion to the injured lung.** Hypoxemia severity maps with respect to perfusion gradient and percent change in vascular resistance within the injured compartment of the lower lung zone. Top row shows contours of the ratio of arterial oxygen tension to inspired oxygen fraction ($P_aO_2$:$F_iO_2$). Bottom row shows contours of the shunt fraction ($F_{shu}$). Columns represent different levels of the fraction of lung injured ($F_{inj}$).

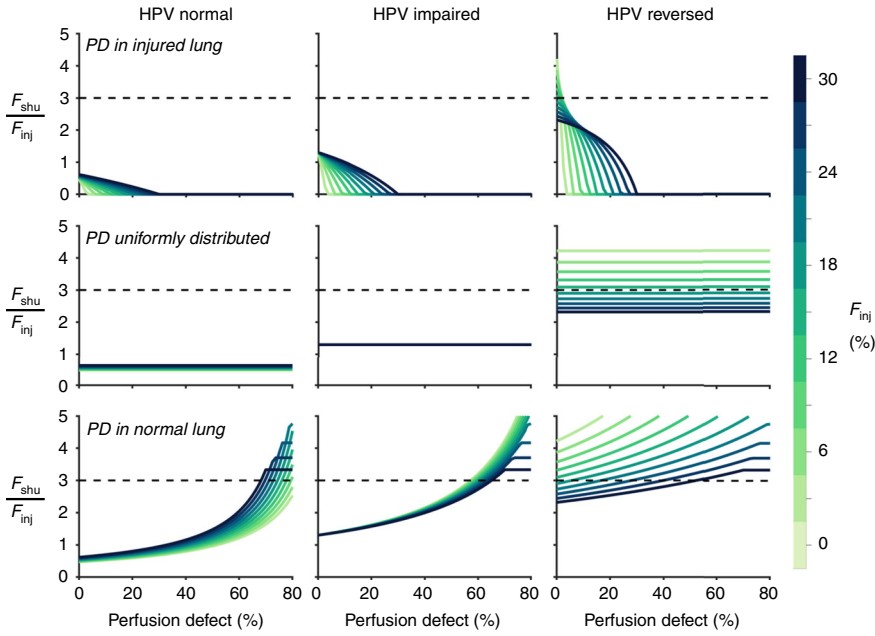

**Fig. 4 Role of perfusion defects in injured and noninjured lung regions.** Influence of perfusion defect (PD) on the ratio of shunt fraction ($F_{shu}$) to injured fraction ($F_{inj}$). Color reflects the fraction of injured lung, concentrated in the lower lung zone. Columns represent types of alterations to hypoxic pulmonary vasoconstriction (HPV). Rows represent types of perfusion defect distributions. Baseline perfusion gradient was 30%, and reversed HPV was modeled with 72% reduction of vascular resistance in injured regions. Dashed lines indicate $F_{shu}$:$F_{inj}$ ratio of 3. Note that $F_{inj} = 0\%$ is not explicitly shown.

perfusion defect (concentrated in the noninjured compartments) also reduces the amount of vasodilation required to produce $F_{shu}$: $F_{inj} \geq 3$ in the reversed HPV model. Nonetheless some degree of vasodilation is required to explain $F_{shu}$:$F_{inj} \geq 3$ for any perfusion defects affecting <60% of the lung.

**Ventilation-perfusion mismatching can explain hypoxemia.** The role of ventilation-perfusion mismatching in the noninjured-perfused lung is characterized by its net effect on venous admixture in blood returned from only the noninjured lung compartments. Figure 5 demonstrates the amount of venous

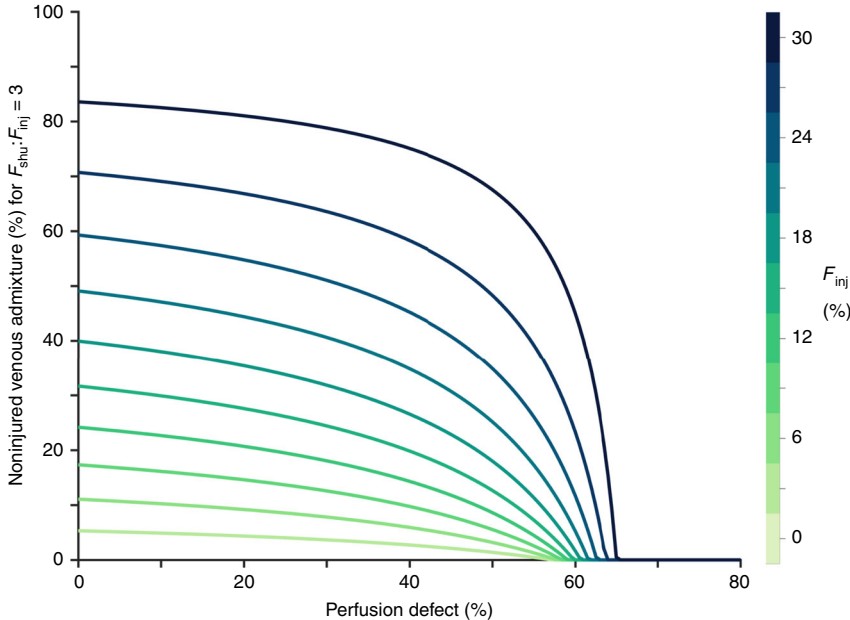

**Fig. 5 Contribution of ventilation-perfusion mismatch in the noninjured lung.** Each line represents the fraction of venous admixture in perfusion from the noninjured lung required to produce a ratio of total shunt fraction ($F_{shu}$) to injured fraction ($F_{inj}$) of 3, assuming complete shunt in the injured lung compartments, and assuming impairment of hypoxic pulmonary vasoconstriction everywhere in the lung. Color indicates the fraction of injured lung ($F_{inj}$). Baseline perfusion gradient was 30%, and injury was focused in the lower lung zone. Note that $F_{inj} = 0\%$ is not explicitly shown.

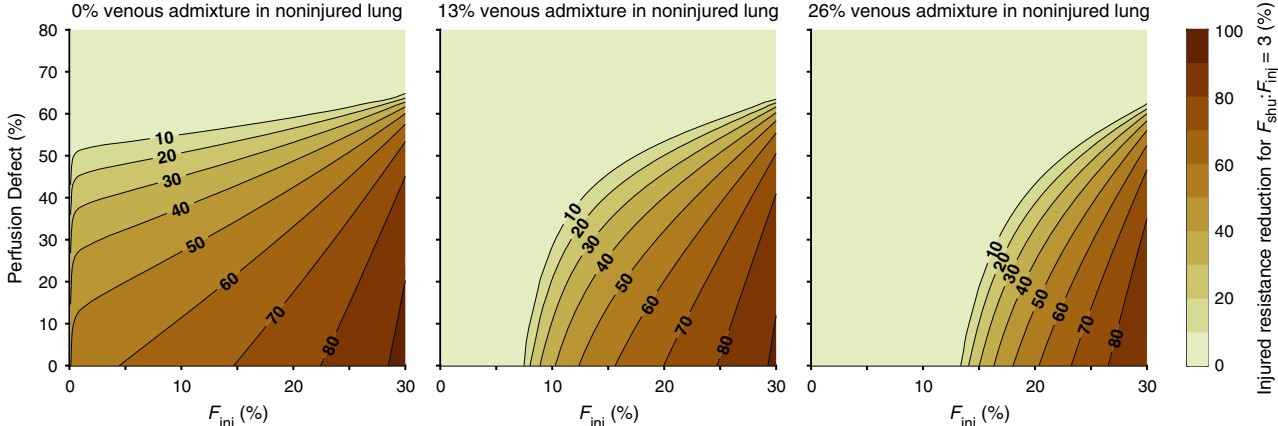

**Fig. 6 Interplay between perfusion defect and altered hypoxic pulmonary vasoconstriction.** Color indicates the amount of vasodilation required to produce a ratio of shunt fraction ($F_{shu}$) to injured fraction ($F_{inj}$) of 3, for a given fraction of injured lung ($F_{inj}$) and fraction of noninjured lung with perfusion defect. Baseline perfusion gradient was 30%, and injury was focused in the lower lung zone.

admixture required to produce $F_{shu}:F_{inj} \geq 3$, assuming impaired HPV everywhere in the model. As shown in Fig. 4, perfusion defect greater than 60% is sufficient for $F_{shu}:F_{inj} \geq 3$ without any vasodilation. Less perfusion defect is required if the noninjured lung produces venous admixture. For example, $F_{shu} = 54\%$ and $F_{inj} = 18\%$ is possible without HPV reversal and with <40% perfusion defect when accompanied by 30–40% venous admixture resulting from poor ventilation-perfusion matching in the noninjured lung.

Figure 6 shows the amount of vasodilation required in the HPV reversed model. Similar to the result shown in Fig. 3, up to 70% reduction in vascular resistance of the injured compartment is required for $F_{shu}:F_{inj} \geq 3$ when there is no ventilation-perfusion mismatching in the noninjured-perfused lung. The required amount of vasodilation is reduced as the size of the perfusion defect increases, and as the amount of noninjured venous

admixture increases. For example, if $F_{inj} = 18\%$ and pulmonary embolism results in 30% perfusion defect in the noninjured lung, then $F_{shu}:F_{inj} \geq 3$ requires 59% reduced resistance with normal ventilation-perfusion matching in the noninjured lung, but only 19.6% reduced resistance with 26% venous admixture from noninjured ventilation-perfusion mismatch.

**Response to inspired oxygen.** The predicted response of the model with 17% $F_{inj}$ to supplemental oxygen is shown in Fig. 7 for several possible scenarios including altered HPV in the injured lung, extensive perfusion defect in the noninjured lung, and venous admixture caused by ventilation-perfusion mismatching in the noninjured lung. In all scenarios, increasing $F_iO_2$ results in increased oxygenation assessed by either arterial oxygen tension or saturation. However, increasing $F_iO_2$ does not improve

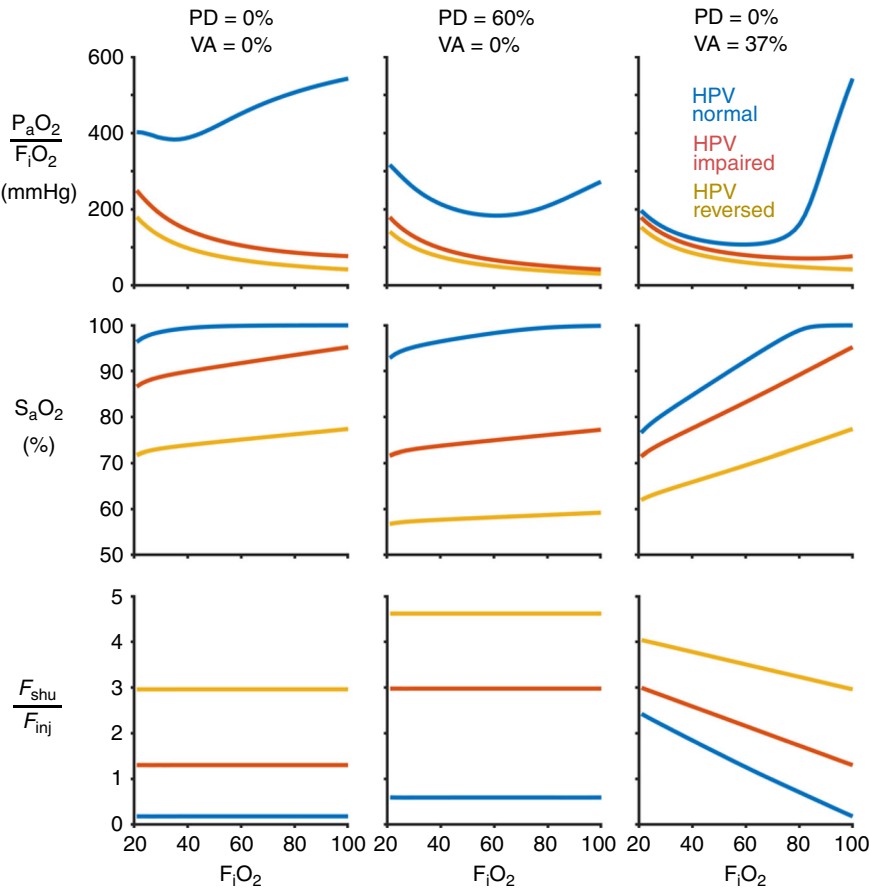

**Fig. 7 Predicted response to increased fraction of inspired oxygen ($F_iO_2$).** Colors correspond to alterations in hypoxic pulmonary vasoconstriction (HPV) within the injured lung compartments (blue: normal; red: impaired; yellow: reversed). Rows correspond to the ratio of arterial oxygen tension to inspired oxygen fraction ($P_aO_2$:$F_iO_2$), arterial oxygen saturation of hemoglobin ($S_aO_2$), and ratio of shunt fraction to injured fraction ($F_{shu}$:$F_{inj}$). Columns correspond to three different cases of perfusion defect in the noninjured lung (PD) and venous admixture owing to ventilation-perfusion mismatching in the noninjured lung at 21% $F_iO_2$ (VA). In all cases, the injured fraction ($F_{inj}$) was 17%, baseline perfusion gradient was 30%, and reversed HPV was modeled with 72% reduction of vascular resistance in injured regions.

the $P_aO_2$:$F_iO_2$ ratio until the mixed arterial blood is fully saturated, which does not occur even at 100% $F_iO_2$ for any model with abnormal HPV. Note that the model of ventilation-perfusion mismatching is the only model for which the ratio $F_{shu}$:$F_{inj}$ is sensitive to changes in $F_iO_2$, and is also the only model for which $F_{shu}$:$F_{inj} > 2$ with normal HPV function.

## Discussion

Our analysis based on a mathematical model of perfusion in normal and shunted compartments suggests several possible explanations for the severe hypoxemia observed in patients with early-stage COVID-19. Despite a relatively small fraction of either poorly aerated or nonaerated injured lung, our model predicts that calculated shunt fractions in excess of three times the injured fraction can be explained by (1) extensive perfusion defect, (2) perfusion defect combined with ventilation-perfusion mismatching in the noninjured lung, or (3) hyperperfusion of the small injured fraction, with up to threefold increases in regional perfusion to the afflicted regions. Although perfusion defects exacerbate pulmonary venous admixture, without perfusion defect affecting >60% of the noninjured lung, some degree of either injured vasodilation or noninjured ventilation-perfusion mismatch is required to explain ratios of shunt fraction to injured fraction >3.

An early stage of COVID-19 characterized by severe hypoxemia but relatively minimal parenchymal involvement[1,3] could be

recapitulated in this model with dramatic reductions to vascular resistance in the injured regions. To replicate the reported values for $F_{shu}$ of 50% and $F_{shu}$:$F_{inj}$ of 3 (implying $F_{inj}$ of 17%), our model without perfusion defect or ventilation-perfusion mismatching requires reductions in injured resistance of 60% to 70%, depending on the baseline perfusion gradient (Fig. 3). Approximating vascular resistance using the Hagen–Poiseuille equation, this change in resistance corresponds to an increase in vascular diameter of 26–35%. Whether this degree of vasodilation from baseline is physiologically plausible seems unlikely, given that maximal vasodilation using inhaled nitric oxide may decrease total pulmonary vascular resistance (PVR) by up to 50%[19,20]. Although speculative, it may be possible that COVID-19 interferes with the HPV feedback mechanism in such a way that pulmonary arterioles do not constrict, and may even dilate, in injured lung regions in which there is little or no oxygen transport into the blood[10,11,13,21,22]. Other evidence of vasodilation owing to COVID-19 includes recent discovery of cardiovascular complications reminiscent of vasodilatory shock and Kawasaki disease[23], which is associated with weakened walls of the coronary artery.

Another factor that may contribute to high $F_{shu}$:$F_{inj}$ is reduced perfusion in well-aerated lung regions due to positive pressure ventilation and application of positive end-expiratory pressure. The reference value of $F_{shu}$:$F_{inj} = 3$ was obtained in patients recently admitted to intensive care and placed on mechanical

ventilators[1], although the levels of positive end-expiratory pressure and driving pressure were not reported. We may speculate that the degree of vasodilation required to explain hypoxemia may be lower in patients not receiving positive pressure ventilation. It is uncertain to what degree the ratio $F_{shu}$:$F_{inj}$ may vary in nonventilated patients compared to this reported value, as our modeling assumptions result from circumstantial evidence and small case series. A recent study using injected microbubbles suggests that vasodilation occurs in some mechanically ventilated patients with COVID-19 and is correlated with $P_aO_2$:$F_iO_2$, however the sample size is small and the measurement technique is nonspecific regarding the location of vascular enlargement and the possibility of intracardiac shunt[11].

Without vasodilation in the model, impairment of HPV alone cannot reproduce the same extreme values of $F_{shu}$:$F_{inj}$ > 2. Instead, the $F_{shu}$:$F_{inj}$ ratio in the impaired HPV model is limited by the magnitude of the baseline perfusion gradient, particularly when the injured region also receives more baseline perfusion (see Fig. 2). The value of $F_{shu}$:$F_{inj}$ = 1.3 reported for ARDS[1,9] is well-matched in our model with a moderate baseline perfusion gradient of 30%, impairment of HPV, and injury focused in the lower compartment (see Fig. 2). This suggests that HPV impairment (e.g., owing to sedatives or anesthetic agents with vasodilating effects) and prevalence of derecruitment in the gravitationally dependent lung (typically dorsal regions in a supine patient) are plausible factors contributing to the observed $F_{shu}$:$F_{inj}$ ratio of 1.3.

In COVID-19, the lower left and lower right lobes are most commonly afflicted according to radiographic abnormalities[5,24], and these are typically the gravitationally dependent regions of the lung in either upright or supine positioning. However, even with an extreme baseline perfusion gradient of 100% (corresponding to 0-1/3-2/3 distribution), $F_{shu}$:$F_{inj}$ in the model is still limited to 2 at most. For example, in Fig. 3, $F_{shu}$:$F_{inj}$ does not exceed 3 even at 100% baseline perfusion gradient until the resistance reduction is 40% for $F_{inj}$ = 10%, or 55% for $F_{inj}$ = 20%. Therefore it appears unlikely that the degree of pulmonary shunt reported in COVID-19 patients ($F_{shu}$ = 50% and $F_{shu}$:$F_{inj}$ = 3) could occur without a substantial degree of vasodilation and hyperperfusion in the small fraction of injured lung, if one also assumes there is negligible venous admixture from other mechanisms producing ventilation-perfusion mismatching.

Coagulation and thrombosis have been identified as prominent symptoms of COVID-19, and in many cases are associated with mortality owing to stroke, myocardial infarction, or pulmonary embolism[14,15,25]. Clinical markers of coagulopathy in COVID-19 are higher for nonsurvivors compared with survivors, and increase over time in nonsurvivors, but are relatively similar upon admission[26]. A retrospective CT study reported pulmonary embolism diagnosed in 22% of COVID-19 patients, with half of those diagnoses made upon admission in the emergency department[27]. Thus, pulmonary embolism, microemboli, and ventilation-perfusion mismatching could also explain hypoxemia in early COVID-19[15]. In our model, heterogeneity of regional ventilation:perfusion ratios is not explicitly considered, but rather the net effects of ventilation-perfusion mismatching in the noninjured compartment are represented as venous admixture specifically from the noninjured compartment. Ratios of $F_{shu}$:$F_{inj}$ ≥ 3 were not observed without either extensive perfusion defect (affecting >60% of the noninjured lung), substantial reductions of vascular resistance in the shunt compartments, substantial venous admixture throughout the noninjured lung, or some combination thereof (see Figs. 4–6).

Massive pulmonary embolism is often accompanied by dyspnea and chest pain[28], but is also commonly silent[29]. With mild or moderate pulmonary embolism, ventilation-perfusion mismatching in noninjured regions and/or vasodilation in injured regions remain necessary to explain $F_{shu}$:$F_{inj}$ ≥ 3. A case report using dual-energy CT demonstrated no obvious indications of pulmonary emboli in the well-aerated lung, but rather vasodilation of pulmonary arteries and hyperperfusion adjacent to infected regions, concluding that a likely explanation for hypoxemia involves "intrapulmonary shunting towards areas where gas exchange is impaired"[10]. Another case report found pulmonary embolism as well as vasodilation in the regions of ground glass opacification[21]. Vascular enlargement has been a common radiographical finding[22]. A recent study highlighted the prevalence of perfusion defects affecting large lung fractions in mechanically ventilated patients with severe COVID-19, as well as dilation of peripheral vessels[30]. Thus, there is strong clinical evidence of vascular dysregulation in the lungs, and there may be potential for poorly aerated injured regions to exhibit at least impairment of HPV if not vasodilation.

Compounding the effects of large perfusion defects with abnormal HPV, a given ratio of $F_{shu}$:$F_{inj}$ may be obtained at a lower level of vasodilation in the injured region (see Fig. 6). A recent study of COVID-19 patients requiring mechanical ventilation reported dead space fractions (i.e., ratio of dead space volume to tidal volume) as high as 45% at the time of intubation[4]. Although this cohort may represent a later stage of COVID-19, these findings support the notion that disease progression is accompanied by perfusion redistribution away from aerated regions. If thrombotic pulmonary emboli occur during the early stages of COVID-19 as well, this could amplify the apparent $F_{shu}$ and hypoxemia. In cases of increased physiologic dead space, ratios of $F_{shu}$:$F_{inj}$ = 3 may occur with lower, more plausible reductions of resistance in injured regions (30 to 50%), compared with the 60–70% reduction required without considering any physiologic dead space (see Fig. 6). The required amount of vasodilation is even less when there is non-negligible venous admixture from the noninjured lung, which can result from maldistributed ventilation and perfusion associated with pulmonary embolism[17,31]. Thus, it seems plausible that microemboli and ventilation-perfusion mismatching can explain COVID-19 hypoxemia with or without hyperperfusion of the small fraction of injured lung. Responsiveness to increased inspired oxygen may help identify the relative contributions of pulmonary shunt and ventilation-perfusion mismatching (see Fig. 7)[32]. Another factor that may contribute to systemic hypoxia is increased oxygen uptake by lung tissues, which may account for up to 20% of total oxygen metabolism in patients with lung injury compared with only 5% at baseline[33].

The purpose of the model was to quantitatively assess the plausibility of the hypothesis that severe hypoxemia in early COVID-19 is the result of hyperperfusion within a small amount of injured lung. Three mechanisms were considered: (1) alterations to hypoxic pulmonary vasoconstriction, (2) thrombosis-mediated perfusion defects, and (3) ventilation-perfusion mismatching in the noninjured lung. The model demonstrates that vasodilation of injured, unoxygenated regions appears to be a plausible yet unnecessary explanation for the reported severity of hypoxemia in early COVID-19, particularly when accompanied by amplification of venous admixture due to thrombotic pulmonary emboli and/or ventilation-perfusion mismatching throughout the noninjured lung. Hypoxemia in early COVID-19 may be explained by relatively small changes in all three factors simultaneously, or by larger changes in only one or two factors, and this should be further investigated.

## Methods

**Model description.** The lung model was partitioned into 12 compartments (Fig. 1), representing three different height levels each partitioned into four

compartments: injured-perfused, normal-perfused, injured-nonperfused, and normal-nonperfused. The three height levels with different gravitational potentials loosely correspond to West zones[34]. The normal, shunt, and dead space compartments correspond to those of the Riley model[35,36]. Perfused compartments received deoxygenated mixed venous blood and returned end-capillary blood with oxygen content determined by injury severity. The model described time-averaged gas exchange, i.e., neglecting within-breath and within-beat fluctuations. Normal lung compartments had normal oxygen equilibration such that end-capillary oxygen tension ($P_cO_2$) equilibrated with alveolar oxygen tension ($P_AO_2$). Injured lung compartments had limited or zero oxygen equilibration such that $P_cO_2$ was either equal to mixed venous oxygen tension ($P_vO_2$) or a weighted average of $P_vO_2$ and $P_AO_2$:

$$P_cO_2 = P_vO_2 + B \cdot (P_AO_2 - P_vO_2) \tag{1}$$

Note that in the normal lung compartments, $B$ was assumed to have a value of 1. A value of 0 for $B$ corresponds to a complete shunt with no oxygen equilibration. Oxygen tensions were assumed to be $P_vO_2 = 40$ mmHg and $P_AO_2 = 100$ mmHg, representing patients upon admission without supplemental oxygen. These values result from the alveolar gas equation, with 21% inspired oxygen, 47 mmHg water vapor pressure at 37 C, 40 mmHg arterial carbon dioxide tension, and 0.8 respiratory quotient:

$$P_AO_2 = 0.21 \cdot (760\,\text{mmHg} - 47\,\text{mmHg}) - \frac{40\,\text{mmHg}}{0.8} \tag{2}$$

Perfusion distribution in the model reflected the relative vascular resistance in each compartment. First, baseline resistances ($R_{bas}$) were determined to establish a baseline perfusion gradient in three equal-sized normal compartments (i.e., in the absence of any injury). Baseline perfusion gradient was defined as half the range of perfusion across all height levels divided by the average. Baseline vascular resistance ($R_{bas}$) at each height level ($h$), relative to baseline total $PVR_{bas}$, was determined as follows:

$$R_{bas}(h) = PVR_{bas} \frac{Q_{tot}}{Q_{bas}(h)} \tag{3}$$

where $Q_{tot}$ is total pulmonary perfusion (i.e., cardiac output), and $Q_{bas}$ is baseline perfusion at each height level. Vascular resistance was then adjusted in injured regions to reflect possible abnormalities arising in COVID-19. Three types of modification were examined: (1) normal HPV function increased resistance exponentially in regions with low $P_cO_2$[18]; (2) impaired HPV function produced no change in resistance; and (3) reversed HPV-reduced resistance by a factor $0 < K < 1$ regardless of oxygenation. The following equations were used:

$$\frac{R_{inj}(h)}{R_{bas}(h)} = \begin{cases} 1 + 100e^{-P_cO_2/10} & \text{HPV normal} \\ 1 & \text{HPV impaired} \\ K & \text{HPV reversed} \end{cases} \tag{4}$$

where $R_{inj}$ is resistance of the injured compartment. Finally, a fraction of alveolar dead space with perfusion defect was defined for each injured ($F_{pdi}$) and noninjured ($F_{pdn}$) compartment at each height level, corresponding to infinite resistance and zero perfusion. Following these modifications and now accounting for injured and dead space partitions at each height level with altered vascular resistance, total PVR was computed by the parallel combination of compartmental resistances.

$$\frac{1}{PVR} = \sum_h \left[ \left( \frac{F_{inj}(h) \cdot (1 - F_{pdi}(h))}{R_{inj}(h)} \right) + \left( \frac{(1 - F_{inj}(h))(1 - F_{pdn}(h))}{R_{bas}(h)} \right) \right] \tag{5}$$

Perfusion to each $n$th compartment was then allocated in inverse proportion to compartmental resistance.

$$\frac{Q_n}{Q_{tot}} = PVR \cdot \left( \frac{F_n}{R_n} \right) \tag{6}$$

where $R_n$ is the resistance of the $n$th compartment and $F_n$ is the fraction of the total lung represented by that compartment. End-capillary oxygen content ($C_cO_2$) was computed based on $P_cO_2$ for each compartment:

$$C_cO_2 = 1.34 \cdot [\text{Hb}] \cdot S_cO_2 + 0.0031 \cdot P_cO_2 \tag{7}$$

where [Hb] is hemoglobin concentration and $S_cO_2$ is oxygen saturation of hemoglobin determined by the Severinghaus equation fit to the oxygen-hemoglobin dissociation curve[37]:

$$S_cO_2 = \left( 1 + \frac{23400}{P_cO_2^3 + 150 \cdot P_cO_2} \right)^{-1} \tag{8}$$

Mixed venous oxygen content ($C_vO_2$) was calculated in the same manner. Mixed arterial oxygen content ($C_aO_2$) was then determined as a perfusion-weighted average of compartmental end-capillary oxygen contents.

$$C_aO_2 = \sum_n (C_cO_2)_n \cdot \frac{Q_n}{Q_{tot}} \tag{9}$$

Shunt fraction ($F_{shu}$) was defined as the ratio of unoxygenated to total blood flow:

$$F_{shu} = \frac{C_{c*}O_2 - C_aO_2}{C_{c*}O_2 - C_vO_2} \tag{10}$$

where $C_{c*}O_2$ represents an ideal noninjured compartment with end-capillary oxygen tension equal to alveolar oxygen tension.

**Simulations and outcomes**. Measured outcomes included $F_{shu}$, ratio $F_{shu}{:}F_{inj}$, and ratio $P_aO_2{:}F_iO_2$. Simulations were conducted over a range of $F_{inj}$ from 0 to 30%. For example, 15% of the lower lung zone injured reflects an overall $F_{inj}$ of 5%, because the lower zone represents 1/3 of the total lung. For simplicity, the extent of injury in each simulation was restricted to only one height zone: lower, middle, or upper. Baseline perfusion gradient varied between 0 and 100% representing a range of perfusion heterogeneity from uniform with 1/3-1/3- distribution at 0% gradient to 0-1/3-2/3 distribution at 100% gradient. Perfusion defect extent was simulated between 0 and 80% of the lung, uniformly distributed or concentrated within either the injured or noninjured compartments. Mixed venous oxygen tension was varied between 20 and 100 mmHg, representing a wide range of possibilities encompassing hypoxia, normal physiology, and venovenous extracorporeal membrane oxygenation. Additional simulations representing critical care patients used increased $F_iO_2$ between 21 and 100%, and increased $F_{shu}$ between 0 and 80%. The degree of impaired oxygen equilibration in the injured compartment varied between 0 and 20%, where 0% represents complete shunt and 100% represents complete equilibration with alveolar gas. Ventilation-perfusion mismatching in the noninjured lung was simulated by varying $B$ in Eq. (1) to produce a specified level of venous admixture at 21% $F_iO_2$. As hypoxemia caused by ventilation-perfusion mismatching is generally responsive to increased $F_iO_2$[32], the simulated venous admixture was linearly reduced from the specified value to zero as the $F_iO_2$ increased from 21% to 100%.

**Reporting summary**. Further information on research design is available in the Nature Research Reporting Summary linked to this article.

## Data availability
Data sharing not applicable to this article as no data sets were generated or analyzed during the current study.

## Code availability
A Matlab script for evaluating the mathematical model described herein is available in the Supplementary Information.

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

## Acknowledgements

This study was supported by National Heart, Lung, and Blood Institute Grants U01-HL-139466 and R01-HL-142702.

## Author contributions

J.H., V.M., J.H.T.B., and B.S. conceived and designed research. J.H. performed experiments and analyzed data. J.H., V.M., J.H.T.B., and B.S. interpreted results of experiments, drafted manuscript, edited and revised manuscript, and approved final version of manuscript.

## Competing interests

The authors declare no competing interests.
