## [Peer Review File · Nature Communications]

Reviewers' Comments:

Reviewer #1:

Remarks to the Author:

The authors present an implementation of the Riley model (which is not quoted) adding the gravity effect (Page 12 line 249-252: "Normal lung compartments had normal oxygen diffusion such that end capillary oxygen tension (PcO₂) equilibrated with alveolar oxygen tension (PAO₂). Injured lung compartments had limited or zero oxygen diffusion such that PcO₂ was either equal to mixed venous oxygen tension (PvO₂) or a weighted average of PvO₂ and PAO₂"). This model has two lung compartments: one with a Va/Q of 1 and one of a Va/Q of zero. Afterwards the authors compute the necessary change in resistance/vessel diameter to shunt blood in the non-exchanging lung regions to obtain the shunt values observed in Covid-19 patients. This ignores the effect of Va/Q mismatch, which may be negligible at FiO₂ near to 1 but the authors choose an FiO₂ of 0.21 (Page 13 - line 254-258: "Oxygen tensions were assumed to be PvO₂ = 40 mmHg and PAO₂ = 100 mmHg, representing patients upon admission without supplemental oxygen. These values result from the alveolar gas equation, with 21% inspired oxygen, 47 mmHg water vapor pressure at 37 C, 40 mmHg arterial carbon dioxide tension, and 0.8 respiratory quotient"). Ignoring the effect of Va/Q mismatch makes the model unreliable in a disease where the pathogenic mechanism includes endothelial insult with pulmonary micro thrombosis and pulmonary infarctions. Of note, in the proposed model would give a perfect oxygenation in case of massive pulmonary embolism.

Reviewer #2:

Remarks to the Author:

Herrmann et al. describe a mathematical model to better understand if hyperperfusion can explain hypoxemia with limited non-aerated lung tissue in patients with COVID-19 infection.

The aim of the study is clear given that there is tremendous discussion on this topic among clinicians. I welcome the alternative approach to the topic. My main concern is that the model might not be reflective of any clinical condition I've encountered so far. I would be interested to see how the authors would explain a mean % non-aerated lung tissue in ARDS of 25-35% (<https://www.nejm.org/doi/full/10.1056/NEJMoa052052>) with a PaO₂/FiO₂ of 200. According to the model; this would also require severe vasodilatation in ARDS, something that is not observed. This flags that the assumptions might be wrong and require additional attention.

Introduction: the very existence of the L phenotype patients is rather speculative and as of yet, not supported by empirical data. Therefore, the authors might change the tone of the introduction and focus more on the fact that some patients have minor parenchymal involvement, yet severe hypoxemia.

Introduction: "If one assumes that ground-glass opacification represents lung that is nonventilated" - this hypothesis is probably false as groundglass typically is -500 to -200 HU and represents poorly aerated lung tissue that probably received some ventilation.

Model assumptions: the model currently has three compartments with a gravitational effect, ventilated and non-ventilated units and differential degrees of perfusion of the non-ventilated units. I would argue that there is a bit more complexity to the actual situation:

- Ground glass probably is ventilated, although poorly, which would decrease the amount of calculated shunt.
- Perfusion defects were not seen throughout the poorly aerated lung tissue, but only in part of it, which would also decrease the amount of calculated shunt.
- The model neglects dead-space, how could this contribute to the problem?
- The model neglects decreased diffusion capacity, which might contribute in poorly-aerated lung

tissue and increases the effects of low V/Q mismatch.

- I would be interested in increasing the area of Finj to higher values (for example 70%) to see if the model does explain values that we frequently encounter in clinical practice.
- The values from Gattinoni AJRCCM 2020 serve as target for the model, however, these were obtained in intubated and mechanically ventilated patients. Could the positive pressure (resulting in decreased perfusion of well-aerated lung tissue) explain some of the problems with modelling reality?

Model outcomes:

- I'm surprised by the relatively high PaO₂/FiO₂ values (all > 200mmHg) despite shunt fractions of >40%. This is very surprising as we encounter lower PaO₂/FiO₂ in clinic with shunt fractions below 40%. I'm unsure why this occurs, but might signal a fundamental mistake in the model.
- The model has to be pushed to extremes to meet the clinical observations, possibly violating several assumptions, as described above.
- I cannot wrap my head around figure 3; the right panels indicate a shunt fraction of >70% but with a PaO₂/FiO₂ of <200mmHg. That is completely out of line with clinical observation. This might be influenced by the selected fio₂ = 21% in the model. Consider also modelling 50% and 100% fio₂ for more clinical comparison.
- Please consider using several clinical cases that have been published of COVID19 as well as other types of respiratory failure to test the predictions of the model.

Discussion:

- "The Type L early stage of COVID-19 described by Gattinoni et al. characterized by severe hypoxia but relatively normal lung compliance, cannot be recapitulated in this model without dramatic reductions to vascular resistance in the injured regions" - please remember that normal compliance is not modelled; it's the amount of non-aerated lung tissue. This is not necessarily related to compliance.
- "but rather vasodilation of pulmonary arteries and hyperperfusion adjacent to infected regions" - to me this case report contradicts the assumptions of the model as perfusion was decreased within the poorly-aerated region, and hyperperfusion occurred in the borderline area, where supposedly, gas exchange would not be limited.
- What do the authors mean with "The direct implications of this study are furthermore limited to palliative care"

Minor:

- low lung stiffness should be low elastance.
- "in this way is often surprisingly low" remember that this is based on 16 patients with no clear description of patient characteristics and quantitative CT images. Please don't present speculation as fact.

Thank you for developing and sharing the exciting model and I hope the predictions will further improve with some tweaks and added layers of complexity.

Sincerely, Lieuwe Bos

Reviewer #3:

Remarks to the Author:

Thank you for this opportunity to read this manuscript.

This is an interesting quantitative description of the disease in the lungs of COVID patients. The manuscript applies current ideas and uses mathematical models to illustrate that a lack of hypoxic vasoconstriction resulting on hyper-perfusion of shunted regions is a consistent physiological

explanation for the pattern of disease seen in COVID patients. This is a novel and important step in our understanding of the presentation of changes in the lung during COVID.

I only have a small number of concerns with the manuscript and only one which I would consider major (1). These follow, in order of my opinion of importance

1) I missed more information about CO₂. If the authors model is correct, and I believe this is the case. Then the resulting hyper-perfusion of shunted areas should lead to hypo-perfusion of other areas and hence high V/Q or alveolar dead space. The authors comment on p9, 179 that one-third of the lung can be physiologic dead space, however at the top of page 11, they note that Type L is not associated with hypercapnia. An increased dead space should lead to a large gradient between end tidal and arterial PCO₂ levels. If hypercapnia is not present, this means that end tidal values must be very low? This begs the question as to whether normocapnia along with very low end-tidal CO₂ levels is the pattern seen in COVID, at least in type L, and whether this pattern would have been simulated by the authors' mathematical model should CO₂ and dead space compartments have been included in the model simulations.

2) I wonder whether some discussion of ECMO is necessary. It is often a critique of poor ECMO settings that high levels of ECMO oxygenation result in high O₂ levels in shunted blood and therefore reduce HPV, effectively increasing shunt. Is there anything to be learned which is similar to that presented here.

3) I believe it to be a minor point, but on p 3, line 56, the authors ask the question "What are the limitations on oxygen diffusion in injured lungs that would be compatible with clinical findings?" I am not sure that his question is appropriate, given that diffusion limitation is unlikely and not modelled. There are, of course, end capillary to alveolar O₂ gradients due to shunt and other V/Q problems.

Thanks again for the opportunity to review this interesting work.

REVIEWER #1

COMMENT: The authors present an implementation of the Riley model (which is not quoted) adding the gravity effect (Page 12 line 249-252: “Normal lung compartments had normal oxygen diffusion such that end capillary oxygen tension (PcO₂) equilibrated with alveolar oxygen tension (PAO₂). Injured lung compartments had limited or zero oxygen diffusion such that PcO₂ was either equal to mixed venous oxygen tension (PvO₂) or a weighted average of PvO₂ and PAO₂”).

RESPONSE: Thank you for directing our attention to the Riley model of V/Q matching. We have now included in the methods on line 259 the following references:

Riley RL, Cournand A. ‘Ideal’ alveolar air and the analysis of ventilation-perfusion relationships in the lungs. *Journal of Applied Physiology* 1(12):825-847, 1949.

Riley RL. Development of the three-compartment model for dealing with uneven distribution. In: Pulmonary Gas Exchange. Ventilation, Blood Flow, and Diffusion, edited by West JB. New York: Academic, 1980, vol. 1, p. 67–85.

COMMENT: This model has two lung compartments: one with a V_a/Q of 1 and one of a V_a/Q of zero. Afterwards the authors compute the necessary change in resistance/vessel diameter to shunt blood in the non-exchanging lung regions to obtain the shunt values observed in Covid-19 patients.

RESPONSE: We would like to clarify that we do not explicitly model alveolar ventilation (V_a), but rather we assume a uniform alveolar oxygen tension (PAO_2). Speaking loosely in terms of V_a/Q , it may be appropriate to state that the “normal” compartments reflect $V_a/Q \sim 1$, but the “injured” compartments reflect $1 > V_a/Q \geq 0$ since we do allow partial oxygen equilibration in some instances of the model.

However, we have substantially modified the model in the revision. The model now accommodates partial oxygen equilibration in the “normal” compartments as well as the injured. This was done to represent the net effect of V_a/Q mismatching in the “normal” lung on venous admixture. This means that the “normal” compartments with perfusion defect now represent $V_a/Q \gg 1$, the “normal” compartments without perfusion defect represent $1 \geq V_a/Q \geq 0$, the injured compartments represent $1 > V_a/Q \geq 0$ (same as before), and the injured compartments with perfusion defect represent $V_a/Q \gg 1$ or $V_a/Q \sim 0/0$.

COMMENT: This ignores the effect of V_a/Q mismatch, which may be negligible at FiO_2 near to 1 but the authors choose an FiO_2 of 0.21 (Page 13 - line 254-258: “Oxygen tensions were assumed to be $PvO_2 = 40$ mmHg and $PAO_2 = 100$ mmHg, representing patients upon admission without supplemental oxygen. These values result from the alveolar gas equation, with 21% inspired oxygen, 47 mmHg water vapor pressure at 37 C, 40 mmHg arterial carbon dioxide tension, and 0.8 respiratory quotient”). Ignoring the effect of V_a/Q mismatch makes the model unreliable in a disease where the pathogenic mechanism includes endothelial insult with pulmonary micro thrombosis and pulmonary infarctions. Of note, in the proposed model would give a perfect oxygenation in case of massive pulmonary embolism.

RESPONSE: Thank you, this is an excellent observation. Hypoxemia in the 1st version of this model with perfusion defect *required* the simultaneous presence of a shunt compartment. Since COVID-19 is very often associated with poorly aerated regions in thoracic CT, we believed at first that these assumptions were appropriate for the scope of this investigation. However, it is entirely plausible that hypoxemia in COVID-19 is due at least in part to V_a/Q mismatching even in the “normal” or noninjured lung regions. We

feel this analysis is highly relevant to this study and its interpretation for COVID-19 pathophysiology, and we thank the reviewer for encouraging us to develop the model further in this direction.

In this revision, we have now incorporated Va/Q mismatching in the noninjured compartments to address the blind spot addressed by the reviewer. This has dramatically changed our conclusions and interpretation. Please note the addition of 3 new figures (Figures 4-6), results (lines 120-153), and extended discussion (lines 206-220, 240-243) addressing the role of venous admixture and perfusion defect in the noninjured lung. We believe this is an extremely important perspective that may provide a more plausible explanation for hypoxemia in this disease, especially given the growing appreciation for the extent of COVID-19-associated coagulopathy.

Additionally, in the first submission, we had folded an interpretation of perfusion defects into the baseline gradient, but perhaps this was confusing. In the revision, we have dedicated a variable for the fraction of alveolar dead space, now intended to directly represent perfusion abnormalities caused by pulmonary embolism, separately from the baseline perfusion gradient attributed to gravitational effects and positioning. We hope that the new analysis dedicated specifically to pulmonary embolism will improve the clarity of these interpretations and highlight their importance in this disease model.

REVIEWER #2

COMMENT: Herrmann et al. describe a mathematical model to better understand if hyperperfusion can explain hypoxemia with limited non-aerated lung tissue in patients with COVID-19 infection. The aim of the study is clear given that there is tremendous discussion on this topic among clinicians. I welcome the alternative approach to the topic. My main concern is that the model might not be reflective of any clinical condition I've encountered so far. I would be interested to see how the authors would explain a mean % non-aerated lung tissue in ARDS of 25-35% (<https://www.nejm.org/doi/full/10.1056/NEJMoa052052>) with a PaO₂/FiO₂ of 200. According to the model; this would also require severe vasodilatation in ARDS, something that is not observed. This flags that the assumptions might be wrong and require additional attention.

RESPONSE: Thank you for highlighting these concerns. First, we would like to note that our model was intended to represent spontaneously breathing patients without supplemental oxygen, i.e., status upon admission with 21% oxygen. We have made this explicit on line 77 in the results, line 270 in the methods, and we refer to “early” stages of

the disease in the Introduction (lines 30-39). The clinical condition of these patients may deteriorate as the disease progresses and the patient is transitioned into the intensive care unit.

Second, the values of the PaO₂:FiO₂ ratio we reported would be much lower if we changed the conditions of the model to reflect patients in critical care, i.e., with higher FiO₂ and lower mixed venous oxygen tension (PvO₂). For example, assuming PvO₂ = 40 mmHg and FiO₂ = 21%, the lowest possible PaO₂:FiO₂ ratio with 100% shunt is only 190. In the revised manuscript results (lines 76-80 and 93-100) we note these assumptions and direct the reader to the supplementary material in which we provide additional simulation results with higher FiO₂, lower PvO₂, and higher Finj - conditions more representative of critical care patients. We feel that these conditions provide an important perspective by juxtaposing the early and late stages of this disease and ensuing ARDS, and we thank the reviewer for suggesting these additional simulations.

COMMENT: Introduction: the very existence of the L phenotype patients is rather speculative and as of yet, not supported by empirical data. Therefore, the authors might change the tone of the introduction and focus more on the fact that some patients have minor parenchymal involvement, yet severe hypoxemia.

RESPONSE: Thank you for this suggestion. We would like to note that we had already introduced the concept of the “L phenotype” as controversial in the first submission. We have heavily revised the Introduction and Discussion. Now we do not call attention to the concept of phenotypes, but instead motivate our study by the intriguing findings of case reports and small cohorts representing “early” stages of the disease. We have also removed references to lung compliance, since respiratory mechanics in COVID-19 are both controversial in the literature and beyond the scope of this model. Instead, we now refer to “minor parenchymal involvement” as suggested (e.g., line 167).

COMMENT: Introduction: "If one assumes that ground-glass opacification represents lung that is nonventilated" - this hypothesis is probably false as groundglass typically is -500 to -200 HU and represents poorly aerated lung tissue that probably received some ventilation.

RESPONSE: We changed this phrasing from “nonventilated” to “poorly ventilated” (e.g., line 43).

COMMENT: Model assumptions: the model currently has three compartments with a gravitational effect, ventilated and non-ventilated units and differential degrees of perfusion of the non-ventilated units. I would argue that there is a bit more complexity to the actual situation:

COMMENT: Ground glass probably is ventilated, although poorly, which would decrease the amount of calculated shunt.

COMMENT: The model neglects decreased diffusion capacity, which might contribute in poorly-aerated lung tissue and increases the effects of low V/Q mismatch.

RESPONSE: Thank you for this suggestion. We already used a variable to reflect partial oxygen equilibration in the “injured” lung compartment (instead of complete shunt with zero oxygen equilibration). In our previous analysis, we demonstrated that a nearly complete shunt was *required* to recapitulate the reported values of $F_{shu}:F_{inj} > 3$, assuming no other V/Q mismatching in the “normal” or noninjured lung. Whether or not a complete shunt accurately corresponds to the ground-glass opacities observed in COVID-19 is unknown. The goal of this modelling study was to demonstrate which variables are necessary to explain the unusual observations reported in literature regarding minimal parenchymal involvement with severe hypoxemia.

However, in the revised manuscript, we also now include potential for V/Q mismatching in the noninjured compartments as well as the injured (Figures 4-6). This was done particularly in response to the comments of Reviewer #1, but we feel it is relevant to your comments as well. This modification to the model adds some complexity, but also provides an alternative and perhaps more plausible explanation for hypoxemia.

COMMENT: Perfusion defects were not seen throughout the poorly aerated lung tissue, but only in part of it, which would also decrease the amount of calculated shunt.

COMMENT: The model neglects dead-space, how could this contribute to the problem?

RESPONSE: Thank you for these suggestions. We have now included a dedicated analysis for perfusion defects and alveolar dead space (Figures 4-6, results 120-153, discussion lines 206-220), intended to directly represent perfusion abnormalities caused by pulmonary embolism separately from the baseline perfusion gradient attributed to gravitational effects and positioning. In this new analysis, we allow the perfusion defects to occur either uniformly throughout the model, or preferentially in the normal or injured compartments. As you suggested, the apparent shunt fraction is reduced when perfusion defects occur preferentially in the injured compartment.

Anatomical dead space was not considered in this model. We do not explicitly model the ventilation required to maintain alveolar oxygen tension at 100 mmHg. There is an implicit assumption that a spontaneously breathing patient will compensate for hypoxemia with increased minute ventilation, which is a common observation in ‘silent hypoxia’.

COMMENT: I would be interested in increasing the area of Finj to higher values (for example 70%) to see if the model does explain values that we frequently encounter in clinical practice.

RESPONSE: Thank you for this suggestion. We would like to point out that the model was intended to describe the spontaneously breathing patient without supplemental oxygen. In the revision, we have now included additional simulations as supplementary material with a much wider range of Finj, as well as FiO₂, that may be more representative of mechanically ventilated critical care patients. We refer to this supplementary material in the results on lines 76-80 and 93-100.

COMMENT: The values from Gattinoni AJRCCM 2020 serve as target for the model, however, these were obtained in intubated and mechanically ventilated patients. Could the positive pressure (resulting in decreased perfusion of well-aerated lung tissue) explain some of the problems with modelling reality?

RESPONSE: Thank you - this is a valid criticism of the work. We have added a statement to the Discussion on line 182 to account for this possibility. To reiterate, our goal was to identify the modeling assumptions required to reproduce the hypoxemia observed in early COVID-19 despite minimal parenchymal involvement. Beyond the Gattinoni letter, there are other reports of 'silent hypoxia' in admitted patients, thus it is likely that positive pressure alone is not sufficient to explain this phenomenon. Nonetheless, we acknowledge the role that positive pressure may have in producing the exact values for Fshu:Finj reported by Gattinoni et al.

COMMENT: I'm surprised by the relatively high PaO₂/FiO₂ values (all > 200mmHg) despite shunt fractions of >40%. This is very surprising as we encounter lower PaO₂/FiO₂ in clinic with shunt fractions below 40%. I'm unsure why this occurs, but might signal a fundamental mistake in the model.

RESPONSE: Please see the response to your first comment. The lowest possible PaO₂:FiO₂ ratio with a PvO₂ = 40 mmHg and FiO₂ = 21% is 190. Increasing the FiO₂, or reducing the PvO₂, will produce lower PaO₂:FiO₂ ratios.

COMMENT: The model has to be pushed to extremes to meet the clinical observations, possibly violating several assumptions, as described above.

RESPONSE: We agree that some of the parameter values required to recapitulate clinical observations represent extreme deviations from their normal values. This was expected, given the counterintuitive reports of severe hypoxia and shunt fraction despite minor

parenchymal involvement, and this was the motivation for our quantitative modelling study. We have been careful to ensure transparency about our modelling assumptions, and all of the simulation outcomes are at least *physically possible* if not *physiologically plausible*. In the revised manuscript, the addition of V/Q mismatching as an alternative source of hypoxemia may present a more plausible explanation compared to the extreme vasodilation required in the model without V/Q mismatching in the “normal” lung.

COMMENT: I cannot wrap my head around figure 3; the right panels indicate a shunt fraction of >70% but with a PaO₂/FiO₂ of <200mmHg. That is completely out of line with clinical observation. This might be influenced by the selected fio₂ = 21% in the model. Consider also modelling 50% and 100% fiO₂ for more clinical comparison.

RESPONSE: Yes, precisely! Please refer to our response to your first comment. We have now included simulations with higher FiO₂ in the supplementary material.

COMMENT: Please consider using several clinical cases that have been published of COVID19 as well as other types of respiratory failure to test the predictions of the model.

RESPONSE: To our knowledge, there are no other published data available for COVID-19 describing both gas exchange physiology (i.e., shunt fraction, blood gas tensions) and fraction of lung involvement.

We have included the reference you suggested to corroborate Gattinoni’s references to $F_{shu}:F_{inj} = 1.3$ in ARDS:

Gattinoni, L. et al. Lung recruitment in patients with the acute respiratory distress syndrome. *N Engl J Med* 354, 1775–1786 (2006).

We have also included references with physiologic data for patients with acute pulmonary embolism, which we compare to our new simulations of perfusion defect and V/Q mismatching:

Santolucando, A. et al. Mechanisms of hypoxemia and hypocapnia in pulmonary embolism. *Am J Respir Crit Care Med* 152, 336–347 (1995).

COMMENT: "The Type L early stage of COVID-19 described by Gattinoni et al. characterized by severe hypoxia but relatively normal lung compliance, cannot be recapitulated in this model without dramatic reductions to vascular resistance in the injured regions" - please remember that normal compliance is not modelled; it's the amount of non-aerated lung tissue. This is not necessarily related to compliance.

RESPONSE: We have removed references to lung compliance, which are beyond the scope of this model. We acknowledge that lung compliance is fraught with confounding variables, such as nonlinear strain-stiffening and intra-tidal recruitment.

COMMENT: "but rather vasodilation of pulmonary arteries and hyperperfusion adjacent to infected regions" - to me this case report contradicts the assumptions of the model as perfusion was decreased within the poorly-aerated region, and hyperperfusion occurred in the borderline area, where supposedly, gas exchange would not be limited.

RESPONSE: We acknowledge that perfusion defects may occur within the “injured” compartment, and we have now included this possibility in our model of pulmonary embolism and alveolar dead space. Nonetheless, the primary question addressed in our investigation is whether or not we can explain severe hypoxia despite minimal parenchymal involvement without assuming hyperperfusion of the “injured” lung. The case report demonstrates hyperperfusion of the borderline area, and states “*these perfusion abnormalities, combined with the pulmonary vascular dilation we observed, are suggestive of intrapulmonary shunting toward areas where gas exchange is impaired, resulting in a worsening ventilation-perfusion mismatch and clinical hypoxia*”. We acknowledge that this is only a case report, and await more definitive evidence using DECT in a larger cohort.

COMMENT: What do the authors mean with "The direct implications of this study are furthermore limited to palliative care"

RESPONSE: This statement was ambiguous and has been removed. We also removed an entire paragraph extrapolating the results of this modeling study to potential therapies.

COMMENT: low lung stiffness should be low elastance.

RESPONSE: We have removed mention of compliance and elastance.

COMMENT: "in this way is often surprisingly low" remember that this is based on 16 patients with no clear description of patient characteristics and quantitative CT images. Please don't present speculation as fact.

RESPONSE: Thank you for highlighting this concern. We have removed this language and heavily revised the Introduction, emphasizing that these findings are reported in small cohorts.

COMMENT: Thank you for developing and sharing the exciting model and I hope the predictions will further improve with some tweaks and added layers of complexity.

RESPONSE: Thank you for the positive comment. We believe the strength of the manuscript and confidence in the model predictions have both improved due to the suggested addition of other perfusion abnormalities in the model.

REVIEWER #3

COMMENT: Thank you for this opportunity to read this manuscript. This is an interesting quantitative description of the disease in the lungs of COVID patients. The manuscript applies current ideas and uses mathematical models to illustrate that a lack of hypoxic vasoconstriction resulting on hyper-perfusion of shunted regions is a consistent physiological explanation for the pattern of disease seen in COVID patients. This is a novel and important step in our understanding of the presentation of changes in the lung during COVID.

RESPONSE: We thank the reviewer for their positive comment.

COMMENT: I missed more information about CO₂. If the authors model is correct, and I believe this is the case. Then the resulting hyper-perfusion of shunted areas should lead to hypo-perfusion of other areas and hence high V/Q or alveolar dead space. The authors comment on p9, 179 that one-third of the lung can be physiologic dead space, however at the top of page 11, they note that Type L is not associated with hypercapnia. An increased dead space should lead to a large gradient between end tidal and arterial PCO₂ levels. If hypercapnia is not present, this means that end tidal values must be very low? This begs the question as to whether normocapnia along with very low end-tidal CO₂ levels is the pattern seen in COVID, at least in type L, and whether this pattern would have been simulated by the authors' mathematical model should CO₂ and dead space compartments have been included in the model simulations.

RESPONSE: Thank you, this is an excellent observation. Although there is limited information regarding the spontaneous breathing patterns of admitted patients, there are frequent reports relating early COVID-19 to 'silent hypoxia'. A possible explanation for this finding is that patients increase their resting minute ventilation to compensate for hypoxia due to shunt and/or alveolar dead space (please see references 8 and 9 in the revised manuscript). In this case, even hypocapnia is possible. Nevertheless, due to the lack of available data regarding CO₂, explanations such as these remain speculative so we chose not to include carbon dioxide tensions as a variable and thus assumed normocapnia.

It is possible to incorporate new model variables accounting for alveolar ventilation (anatomic dead space, respiratory rate, tidal volume), CO₂ tensions, various CO₂ reservoirs and buffers in the blood, metabolic CO₂ production rate, and the actual cardiac output (rather than a normalized value). However, this would greatly increase the complexity of the model while contributing relatively little to addressing the current question of why hypoxemia occurs in the presence of minor parenchymal involvement. We agree that the question of carbon dioxide elimination in early COVID-19 warrants further investigation, but we feel that this would require more and better clinical data before such a modelling effort could be properly informed.

COMMENT: I wonder whether some discussion of ECMO is necessary. It is often a critique of poor ECMO settings that high levels of ECMO oxygenation result in high O₂ levels in shunted blood and therefore reduce HPV, effectively increasing shunt. Is there anything to be learned which is similar to that presented here.

RESPONSE: This is an interesting point. What we consider in our study are the consequences for pulmonary perfusion of a fundamental impairment in HPV. However, there are also consequences for pulmonary perfusion in the case that HPV is tricked by inappropriate venous oxygen tension, as can occur in veno-venous ECMO. This makes an interesting contrast which we now allude to on lines 79-80, where we refer to additional simulation results in the supplementary material. Note that this problem is irrelevant during veno-arterial ECMO.

COMMENT: I believe it to be a minor point, but on p 3, line 56, the authors ask the question "What are the limitations on oxygen diffusion in injured lungs that would be compatible with clinical findings?" I am not sure that his question is appropriate, given that diffusion limitation is unlikely and not modelled. There are, of course, end capillary to alveolar O₂ gradients due to shunt and other V/Q problems.

RESPONSE: Thank you for this suggestion. We have rephrased this question in terms of "impaired oxygen diffusion" on line 56, and also clarified on lines 50-51 that diffusion limitation may arise when inflammation and edema thicken the blood-gas barrier. We do not presume to know the state of ventilation and perfusion in the "injured" regions of ground-glass opacification, but rather we explore a range of possibilities by allowing either zero or partial oxygen equilibration.

Reviewers' Comments:

Reviewer #1:

Remarks to the Author:

Manuscript improved, the present version is fine for me.

Reviewer #2:

Remarks to the Author:

The authors have adequately answered most of my questions and have done an excellent revision of the manuscript.

I do have several textual additional points that require attention:

- Citation 3&4 do not present any data supportive of the statement where it is quoted. Citation 2 contradicts the statement, suggesting that it is an uncommon phenomenon (at least 25% non-aerated lung tissue in that study). The authors should clarify that there is "at least a subset of patients with COVID-19 who present with hypoxemia on room air and show minimal non-aerated lung tissue on chest CT imaging but the frequency of this presentation is uncertain and conflicting between studies".
- Both in the introduction and discussion the authors state that: " The value of $F_{shu}:F_{inj} = 1.3$ for typical ARDS reported by Gattinoni et al" - I would like to emphasize that there is no such thing as typical ARDS and the spread of $F_{shu}:F_{inj}$ that is encountered is enormous. This should be acknowledged.
- The authors should emphasize in the introduction that they work on explaining hypoxemia in patients with minimal parenchymal involvement and that these results are only applicable to those cases and that it is uncertain how frequent such pathophysiology occurs, even in COVID19.
- I believe science should not be about the person writing the paper, but about the conclusions and I would like to discourage the frequent use of "author et al." in the text.
- Figure S2&6: y-axis should be adjusted to clinically relevant values, so between 70% and 100%.
- Figure S10 is more suited for the main body of the manuscript as it perfectly illustrates the influence of f_{iO_2} , which is one of the major knobs turned by the physician.
- The conclusion much better suits the clinical picture than in the previous version. The authors should clarify that with about 20% injured lung the hypoxemia may be explained by a relatively small change in all three factors simultaneously OR with a large change in one of the factors and that this should be further investigated.
- A paragraph should be devoted to the fact that the $F_{shu}:F_{inj}$ of 3 is derived from patients undergoing invasive mechanical ventilation and that the application of PEEP and positive inspiratory pressure could mitigate the need for increases in regional perfusion to the afflicted regions to explain hypoxemia. This is now just one sentence at the end of the paragraph in the discussion and fails to capture this point sufficiently for the uninformed reader.
- Pulmonary embolism is generally reserved for large clots in the setting of venothrombotic disease. However, it is increasingly recognized that microthrombosis resulting from an inflammatory response is responsible for part of the perfusion defects seen in COVID19 (as in non-COVID19 related ARDS). Therefore, it might be better to speak about thrombosis mediated perfusion defects than "massive pulmonary embolism".
- In the discussion it should be acknowledged and repeated that the basic assumption of the model (namely low parenchymal involvement combined with $F_{shu}:F_{inj}$ of 3 under spontaneous, negative pressure ventilation with room air) should be investigated, as these assumptions now result from circumstantial evidence and small case series that do not measure all of these factors simultaneously. We might question this assumption because a shunt fraction of 50% would mean that there is little to no response in paO_2 after increasing f_{iO_2} , and a decrease in PaO_2/F_{iO_2} as suggested figure S10, while this is not a problem that is frequently encountered in the clinic. This could be a simple way to evaluate this phenomenon with clinical data.

Thank you for improving the manuscript and good luck with the revisions.

Reviewer #3:

Remarks to the Author:

Thanks again for the opportunity to review this work. My comments have been addressed, although I might have a difference of opinion with regard CO₂ where the authors state that adding CO₂

"would greatly increase the complexity of the model while contributing relatively little to addressing the current question of why hypoxemia occurs in the presence of minor parenchymal involvement."

I agree the complexity would be increased, but understanding the high V/Q or alveolar dead space would, I believe, help understand this physiological system, as I think that it must be the case that a reduction in HPV increases high V/Q regions. The authors are probably correct however that it is the data illustrating end tidal to arterial CO₂ gradient, rather than the model that would help in such understanding, that these data are sparse, and as such no further modelling is warranted.

I have no further comments on this work.

REVIEWER #1

COMMENT: Manuscript improved, the present version is fine for me.

RESPONSE: Thank you.

REVIEWER #2

COMMENT: The authors have adequately answered most of my questions and have done an excellent revision of the manuscript. I do have several textual additional points that require attention:

RESPONSE: Thank you for your constructive review.

COMMENT: Citation 3&4 do not present any data supportive of the statement where it is quoted. Citation 2 contradicts the statement, suggesting that it is an uncommon phenomenon (at least 25% non-aerated lung tissue in that study). The authors should clarify that there is "at least a subset of patients with COVID-19 who present with hypoxemia on room air and show minimal non-aerated lung tissue on chest CT imaging but the frequency of this presentation is uncertain and conflicting between studies".

RESPONSE: We have removed references 3 and 4 from this statement on lines 31-33, and we have changed the phrasing as suggested.

COMMENT: Both in the introduction and discussion the authors state that: " The value of $F_{shu}:F_{inj} = 1.3$ for typical ARDS reported by Gattinoni et al" - I would like to emphasize that there is no such thing as typical ARDS and the spread of $F_{shu}:F_{inj}$ that is encountered is enormous. This should be acknowledged.

RESPONSE: Thank you for this suggestion. We have replaced the original text "... very large compared to typical value of 1.3 for ARDS" by "... large compared to values of 1.25 ± 0.8 reported for ARDS" on line 42 in the Introduction. We have removed "typical" from line 211 in the Discussion.

COMMENT: The authors should emphasize in the introduction that they work on explaining hypoxemia in patients with minimal parenchymal involvement and that these results are only applicable to those cases and that it is uncertain how frequent such pathophysiology occurs, even in COVID19.

RESPONSE: We have amended the first sentence of the Introduction on lines 31-33 as previously suggested to clarify this scope. The scope of the model is reiterated in the Results on lines 79-80, in the Discussion on lines 169-171, lines 180-181, and in the final paragraph on line 274-276. Uncertainty regarding the frequency of presentation is reiterated in the Discussion paragraph on lines 195-207.

COMMENT: I believe science should not be about the person writing the paper, but about the conclusions and I would like to discourage the frequent use of "author et al." in the text.

RESPONSE: We found 4 instances of this usage in the main text (lines 44, 94, 181, 211 and Figure 2 legend) and removed all of them.

COMMENT: Figure S2&6: y-axis should be adjusted to clinically relevant values, so between 70% and 100%.

RESPONSE: We apologize, there was a mistake in the previous versions of these figures. The tick labels for SaO_2 should have ranged 40% to 100% in both Figures S2 and S6 (as it is correctly shown in Figure S10). This range was shown for the sake of completeness and consistency, since some of the values in Figure S6 extended below 70%. However, we agree with the reviewer that any oxygen saturation below 70% represents a critical life threatening condition and the exact value is not clinically relevant. We have reproduced

these figures in the range 70% to 100%, as suggested. We have amended the Figure S6 legend with "Note that only SaO₂ above 70% are shown".

COMMENT: Figure S10 is more suited for the main body of the manuscript as it perfectly illustrates the influence of FiO₂, which is one of the major knobs turned by the physician.

RESPONSE: Although we can agree that it is valuable to show the influence of FiO₂, we believe that Figure S10 is not the best choice for two reasons: (1) it places too much emphasis on the HPV alterations without regard to the alternative explanations provided by V/Q mismatching and perfusion defects; and (2) it extends beyond the well-defined scope of the early patient with minimal parenchymal involvement. Instead, we elected to relocate Figure S11 to the main text (where it is now Figure 7), which captures the salient features of Figure S10 (namely, response to increased FiO₂) while also providing additional perspective beyond just the altered HPV models.

COMMENT: The conclusion much better suits the clinical picture than in the previous version. The authors should clarify that with about 20% injured lung the hypoxemia may be explained by a relatively small change in all three factors simultaneously OR with a large change in one of the factors and that this should be further investigated.

RESPONSE: We agree, and we have added the suggested statement to the conclusion on line 282-284.

COMMENT: A paragraph should be devoted to the fact that the Fshu:Finj of 3 is derived from patients undergoing invasive mechanical ventilation and that the application of PEEP and positive inspiratory pressure could mitigate the need for increases in regional perfusion to the afflicted regions to explain hypoxemia. This is now just one sentence at the end of the paragraph in the discussion and fails to capture this point sufficiently for the uninformed reader.

RESPONSE: We agree that this is an important limitation and we have separated this statement into its own paragraph on lines 195-207.

COMMENT: Pulmonary embolism is generally reserved for large clots in the setting of venothrombotic disease. However, it is increasingly recognized that microthrombosis resulting from an inflammatory response is responsible for part of the perfusion defects seen in COVID19 (as in non-COVID19 related ARDS). Therefore, it might be better to speak about thrombosis mediated perfusion defects than "massive pulmonary embolism".

RESPONSE: Thank you for this suggestion. We have replaced this phrasing throughout the main text and supplemental material.

COMMENT: In the discussion it should be acknowledged and repeated that the basic assumption of the model (namely low parenchymal involvement combined with $F_{shu}:F_{inj}$ of 3 under spontaneous, negative pressure ventilation with room air) should be investigated, as these assumptions now result from circumstantial evidence and small case series that do not measure all of these factors simultaneously. We might question this assumption because a shunt fraction of 50% would mean that there is little to no response in paO_2 after increasing fiO_2 , and a decrease in PaO_2/FiO_2 as suggested figure S10, while this is not a problem that is frequently encountered in the clinic. This could be a simple way to evaluate this phenomenon with clinical data.

RESPONSE: We have added a cautionary statement to the Discussion on lines 201-203 acknowledging that the key assumptions of our study are based on circumstantial evidence and small case series. We have also added a reference to a recent study using injected microbubbles to assess the presence of pulmonary vascular enlargement in COVID-19, which supports the notion that vasodilation occurs but is nonspecific with regard to the location of vasodilation (e.g., within injured vs. noninjured regions of lung).

We agree that hypoxemia caused by pulmonary shunt does not respond well to increased FiO_2 , whereas hypoxemia caused by V/Q mismatching does. Previously, we made note of this caveat in the supplemental material discussing Figure S11, lines 140-144. At 100% FiO_2 , even poorly ventilated regions of the lung ($0 < V/Q < 1$) should be highly oxygenated, and this provides a differentiated response compared to true shunt ($V/Q = 0$) from calculated shunt. Our model did not replicate this behavior in the right panels of the previous Figure S11 because the parameter we controlled was the amount of venous admixture in blood from the noninjured lung. However, a more appropriate model of hypoxemia caused by V/Q mismatching would exhibit reduced venous admixture at 100% FiO_2 . Therefore, we have revised our model in the simplest possible way to account for this: the specified noninjured venous admixture now corresponds only to $FiO_2 = 21\%$, and linearly decreases to 0% as FiO_2 increases to 100%. This has been explained in the Methods on lines 350-354. We note that this change affects only Figures 7 and S11 (previously S11 and S12, respectively), since these are the only figures for which both noninjured venous admixture $> 0\%$ and $FiO_2 > 21\%$ were simulated. Also note that we have moved previous Figure S11 into the main text as Figure 7, in response to your above comment.

REVIEWER #3

COMMENT: Thanks again for the opportunity to review this work. My comments have been addressed, although I might have a difference of opinion with regard CO₂ where the authors state that adding CO₂ "would greatly increase the complexity of the model while contributing relatively little to addressing the current question of why hypoxemia occurs in the presence of minor parenchymal involvement." I agree the complexity would be increased, but understanding the high V/Q or alveolar dead space would, I believe, help understand this physiological system, as I think that it must be the case that a reduction in HPV increases high V/Q regions. The authors are probably correct however that it is the data illustrating end tidal to arterial CO₂ gradient, rather than the model that would help in such understanding, that these data are sparse, and as such no further modelling is warranted. I have no further comments on this work.

RESPONSE: We agree that understanding the prevalence and impact of high V/Q regions would help understand the system in more detail. The perfusion defects now included in the model serve this role, albeit in a limited capacity, only to investigate whether the gross changes in oxygenation may reasonably result from such defects. We completely agree that including CO₂ would provide a much higher degree of fidelity and utility. However, as you acknowledge, we are limited in our ability to compare such predictions with clinical data.